# Wildflower phenological escape differs by continent and spring temperature

Benjamin R. Lee [1,2,3,4] ✉, Tara K. Miller[4], Christoph Rosche[5,6], Yong Yang[7], J. Mason Heberling[1,2], Sara E. Kuebbing[2,8] & Richard B. Primack[4]

Temperate understory plant species are at risk from climate change and anthropogenic threats that include increased deer herbivory, habitat loss, pollinator declines and mismatch, and nutrient pollution. Recent work suggests that spring ephemeral wildflowers may be at additional risk due to phenological mismatch with deciduous canopy trees. The study of this dynamic, commonly referred to as "phenological escape", and its sensitivity to spring temperature is limited to eastern North America. Here, we use herbarium specimens to show that phenological sensitivity to spring temperature is remarkably conserved for understory wildflowers across North America, Europe, and Asia, but that canopy trees in North America are significantly more sensitive to spring temperature compared to in Asia and Europe. We predict that advancing tree phenology will lead to decreasing spring light windows in North America while spring light windows will be maintained or even increase in Asia and Europe in response to projected climate warming.

Temperate deciduous forests are a dominant biome in the northern hemisphere, covering extensive regions in eastern Asia, continental Europe, and eastern North America[1]. They are characterized by winter-deciduous plant species, and the phenology of leaf out and flowering on all three continents is sensitive to variation in average spring temperature[2–5]. Furthermore, recent research has found disparities in phenological sensitivity among plants from different continents[2,6], suggesting that climate change may affect temperate forests differently depending on location.

Previous studies indicate differences in spring phenological sensitivity between woody and herbaceous plants in forests, with direct comparisons available for plants in Asia[3] and North America[5,7]. Evidence in Europe similarly suggests differences in sensitivities between woody and herbaceous plants[8,9]. Herbaceous plants may be less sensitive to spring temperature than trees because the former often overwinter underground and, therefore, may not respond to the same phenological cues as the latter, which have extensive aboveground structures[10,11], but this mechanism has only been studied in North American forests. This potential phenological mismatch is important because spring-active wildflowers in temperate deciduous forests, as well as some woody understory species[12–15], often rely on leafing out before the canopy closes in order to assimilate 40–100% of their annual carbon budget[16]. The success of this strategy to maximize the spring light window, referred to as "phenological escape"[12,17], is directly associated with patterns of growth[10,12,17–19], survival[12,18–20], flowering[17,20], and reproductive output[10,17,21,22]. Recent studies suggest that the duration of spring light windows in eastern North America is likely to be significantly altered with warming climates[7,13], with herbaceous species generally expected to experience shorter spring light windows in the coming decades.

Experimental evidence for such mismatches, however, has generally been limited to comparisons of woody plants located in common garden experiments, which are limited in geographical and temporal extent and have not evaluated mismatch with understory

[1]Section of Botany, Carnegie Museum of Natural History, Pittsburgh, PA, USA. [2]Department of Biological Sciences, University of Pittsburgh, Pittsburgh, PA, USA. [3]Holden Forests and Gardens, Kirtland, OH, USA. [4]Biology Department, Boston University, Boston, MA, USA. [5]Institute of Biology/Geobotany and Botanical Garden, Martin-Luther-University Halle-Wittenberg, Halle, Germany. [6]German Centre for Integrative Biodiversity Research (iDiv), Halle-Jena-Leipzig, Germany. [7]Co-Innovation Center for Sustainable Forestry in Southern China, College of Biology and the Environment, Nanjing Forestry University, 159 Longpan Rd., Nanjing 210037, China. [8]Yale Applied Science Synthesis Program, The Forest School at the School of the Environment, Yale University, 195 Prospect Street, New Haven, CT, USA. ✉e-mail: LeeB@CarnegieMNH.org

species. For example, Zohner and Renner[6,23] have contributed valuable insights into interspecific variation in plant phenology using woody plants growing in European botanical gardens. It is unknown, however, if the variation in woody plant phenology found in common garden experiments are also observed at large scales, for forest trees growing in their natal environment, and across long time periods[24]. It is similarly unknown whether understory herbaceous plant phenological sensitivity varies across large spatial and temporal scales.

Here we assess intercontinental, long-term data on the phenological sensitivity of canopy and understory forest plants and the potential for phenological mismatch across temperate forests in North America, Europe, and Asia. We evaluate the spring phenology of 5522 herbarium specimens collected between 1901 and 2020, representing 22 common tree and 18 common wildflower species found in temperate deciduous forests on three continents (six wildflower species per continent and six, six, and ten tree species in Europe, North America, and Asia, respectively; see full species list in Table S1 and distribution of observations in Fig. S1). Herbarium collections provide data from across large temporal (centuries) and spatial (intercontinental) scales difficult to match with other methods. Following previously validated methods[25,26], we model Leaf Out Date (LOD) of overstory canopy trees and First Flowering Date (FFD) of spring-blooming understory wildflowers (perennial herbaceous species that leaf out at approximately the same time as they flower in the early spring) in response to average spring (March–April) temperature of the collection year and location. This period is chosen because most (> 99%) of the observed phenology occurred during or following this period (Fig. 1a; means and standard deviations are listed in Table S2) and because initial variable analysis identified this temperature variable as the best for predicting tree and wildflower spring phenology. Additionally, owing to the absence of daily weather records for the vast majority of years and sampling locations, especially Asia, we were unable to calculate and analyze finer-resolution daily temperature data for most geographic locations, which would be required for intercontinental comparisons (see Supplementary Information). We use estimated model parameter values to compare phenological sensitivities of trees and wildflowers among continents and to project how spring light window duration will change by the end of the current century based on projected climate trends. Our results suggest that there are differences in the sensitivity of trees (but not wildflowers), with North American trees considerably more sensitive to spring temperature than Asian or European trees. This difference has implications for phenological escape and the future conservation of the species-rich herbaceous layer in temperate deciduous forests as the climate continues to change.

## Results and discussion

We used a hierarchical Bayesian modeling approach to evaluate the relationship between the spring phenology of tree and wildflower species and various climate drivers (see Methods). Following model selection, our final model structure included fixed effects of average spring (March–April) temperature and elevation, as well as species-level random effects. We show continental distributions of spring temperature values in Fig. 1b (means and standard deviations are listed in Table S2). We report estimates for spring temperature sensitivities from the final model structure in the main text. Parameter estimates for elevation sensitivities as well as the model performance of other potential drivers and combinations of drivers are reported in Tables S3 and S4. An extended discussion of model assumptions and limitations is included in the Supplementary Information.

### Sensitivity differences by strata

Tree leaf out phenology (LOD) was substantially more sensitive to average spring temperature in North America (mean = −3.62 days °C⁻¹; 95% credible interval (CI) = [−3.76, −3.49]) than in Europe (mean = −2.79; CI = [−3.27, −2.30]) and Asia (mean = −2.62; CI = [−2.97, −2.26]; Fig. 2). These values are consistent with previously reported phenological sensitivities in North America[7] (−5.5 to −3.3 days °C⁻¹) and Europe[8] (−4.1 to −3.0 days °C⁻¹), as the credible intervals from our results overlap with the reported credible intervals of prior studies. However, the Asian LOD sensitivity was less sensitive than previously reported[27] (−3.50 to −3.03 days °C⁻¹), potentially owing to differences in species selection[28] or model structure. Previously reported sensitivities were determined in separate studies using either observational data[7,8] or long-term observation-based weather station data[27]. The general consistency between our findings suggests that phenology data from herbarium collections are good indicators of patterns in natural systems[29–31], a point supported by a recent study of phenological sensitivity derived from herbaria and from observed citizen science data[32]. These herbarium-based results provide evidence that phenological sensitivity differs across the temperate forest biome (but see ref. 33 for evidence of differences in response to warming and chilling accumulation). To our knowledge, our study is the first to contrast overstory and understory phenology across multiple continents and, therefore, to find differences in phenological sensitivity between trees and forest wildflowers across continents. We recommend future studies explore these differences using alternative approaches and methodologies that focus on the physiological basis for and mechanisms that underlie these patterns.

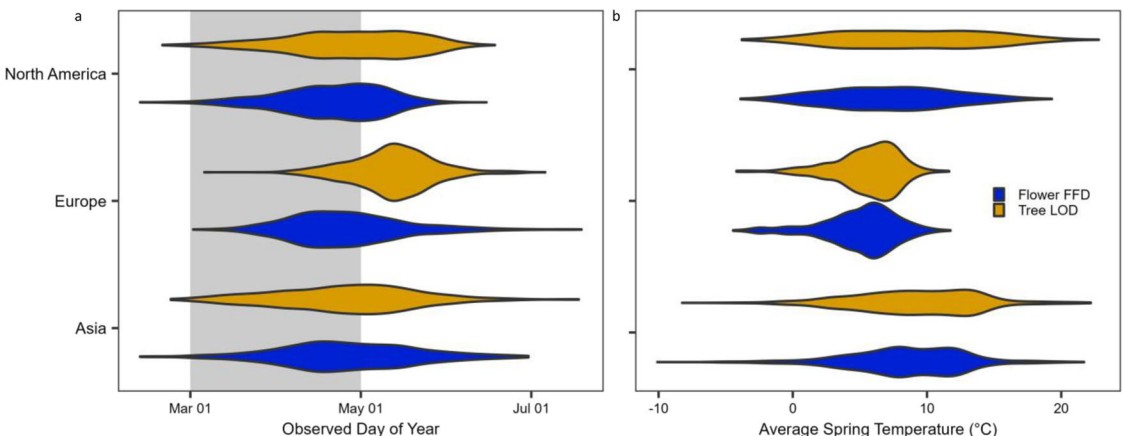

**Fig. 1 | Distributions of observed phenology in the herbarium record by (a) observed date of phenology and (b) average spring (March–April) temperature (°C).** Distributions of observed understory wildflower First Flowering Date (FFD; blue) and canopy tree Leaf-out Date (LOD; gold) in temperate forests in Asia, Europe, and North America. Grey shading behind the violin plots in panel a indicates the March–April period used to model and forecast phenology.

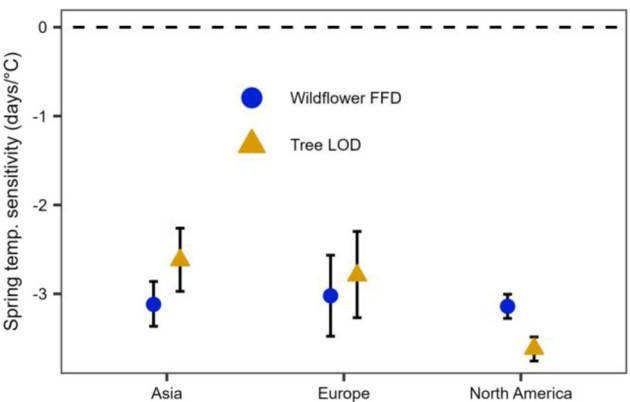

**Fig. 2 | Posterior estimated means and 95% credible intervals for spring temperature sensitivity.** Shapes represent parameter estimates for wildflower First Flower Date (FFD, blue circles; $n = 1418$, 618, and 1060 for Asia, Europe, and North America, respectively) and canopy tree Leaf Out Date (LOD, yellow triangles; $n = 899$, 532, and 995, for Asia, Europe, and North America, respectively). Estimates are considered different from 0 if credible intervals do not overlap the dashed 0 line and are considered different from each other if credible intervals do not overlap.

In contrast to trees, wildflower sensitivity to spring temperature was similar across all three continents and exhibited no strong differences (i.e., overlap in 95% Bayesian credible intervals) among continents (means and 95% credible intervals in brackets: North America = −3.14, [−3.28, −3.00]; Europe = −3.02, [−3.48, −2.56]; Asia = −3.12, [−3.36, −2.86]; Fig. 2). These values are also generally consistent with those reported elsewhere in the literature (i.e., 95% credible intervals overlap with those reported in other studies; −2.2, [−3.7, −0.76] days °C⁻¹ in North America[7] and −3.6, [−4.04, −3.18] days °C⁻¹ in Europe[9]), although we are unaware of any studies that have estimated phenological sensitivity for Asian forest wildflowers in days °C⁻¹. Ge et al.[3] report herbaceous plant sensitivity of −5.71 days per decade in Asia (±7.90 standard deviation; based primarily on long-term observational data), which appears to be roughly consistent with our model results, but the difference in units makes this more speculative than the other comparisons. Discrepancies in mean responses between this study and others may be due in part to different types of data (herbarium specimens versus field observations) and to choice in focal taxa, as temperature sensitivity has been shown to vary widely across taxa[28].

Particularly noticeable in our results was that $r^2$ coefficients of predicted versus observed phenology were much higher in North America (0.70 and 0.76 for wildflower and tree models, respectively) compared to Asian (0.40 and 0.44, respectively) and European models (0.41 and 0.25, respectively). This difference in model performance could be due to the higher interannual variability of spring temperatures in North America[33], leading to greater selective pressure for strong sensitivity to spring temperatures in North American plants. This difference could explain why North American species exhibit higher correlation of phenology with average spring temperatures (Table S4). Alternatively, European and Asian species may have stronger phenological responses to alternative spring forcing windows, winter chilling temperatures, or photoperiod, relative to the March–April temperature period used in this study (see Methods). We think the latter explanation is unlikely, given the strong correlations of phenology with spring temperature across all continents (see Supplementary Information – Justification for March–April Temperature Window).

Herbarium-based phenological models may be improved by accounting for spatial autocorrelation within the dataset. For example, Willems et al.[9] found that including spatial autocorrelation significantly

improved predictability of European herbaceous flowering phenology, even when accounting for multiple drivers of spring phenology. We followed a similar approach as their study and found similar improvements in model performance with the addition of spatial autocorrelation (Tables S3–S4) that had substantial positive effects on $r^2$ values of Asian and European models. However, spatial distributions of specimens differed substantially among continents (see Figs. S2–S4), and these differences could lead to artifacts that make results unreliable to interpret (see Supplementary Information). Therefore, we focus here on results for models without spatial autocorrelation while acknowledging that spatial aggregation of herbarium specimens in Europe and Asia may be partially responsible for the relatively lower $r^2$ values. We encourage other researchers to explore this question further both with our data set and other datasets.

## Climate change and spring light windows
The relative difference between wildflower and tree sensitivity varied substantially among continents, with wildflowers being approximately equally as sensitive to spring temperature as trees in Asia and Europe but substantially less sensitive (i.e., 95% BCI do not overlap) than trees in North America (Fig. 2). Importantly, these differences were driven by changes in tree phenological sensitivities among continents and resulted in different expectations for spring light window duration (i.e., the difference in time between estimated wildflower flowering date and canopy tree leaf out date) on different continents under current climate conditions (Fig. 3), based on modeled leaf out and flowering under a climate scenario derived from average climate conditions from 2009–2018 (Fig. S5).

Interestingly, the time between leaf out and flowering in North America is greater in the north than in the south (Fig. 3c), indicating a greater spring light window duration at higher latitudes. In addition, although there was regional variation in spring light window duration on each continent, there was broad overlap in duration among continents estimated under current environmental conditions (North America light window duration averages 11.7 ± 4.1 s.d. days; European duration averages 14.7 ± 3.0 s.d. days; and Asian duration averages 8.0 ± 4.6 s.d. days). This suggests that, under current climate conditions, wildflowers across all continents experience similar length of spring light windows, but the impact of warming on shrinking or expanding window size will differ among continents with uncertain impacts on wildflower populations.

Still, differences among continents resulted in different projections for how spring light window duration will respond to climate change over the coming century (Fig. 4), which will likely have implications for understory plant demography[7,12,13,16]. We used climate change projections for 2081–2100 (assuming an extreme climate change scenario; Institut Pierre-Simon Laplace CM6A-LR climate model – shared socioeconomic pathway 585; see Methods for more details) to forecast wildflower flowering date and canopy tree leaf-out date for the end of the current century (Fig. S6) and then calculated the difference between forecasted wildflower flowering and tree leaf out phenology for the period of 2081–2100 (future spring light window duration; Fig. S7). To estimate the change in spring light window duration between now and the end of the century (Fig. 4), we subtracted the forecasted future light window duration from the modeled light window duration under current climate conditions.

Dramatic differences in the projected change in spring light window duration emerged, ranging from increasing spring light window durations for wildflowers in Asia, to minimal change in spring light window length in Europe, to decreasing spring light window length in North America. Importantly, these differences in phenological escape trajectories are primarily attributable to differences in phenological sensitivities between overstory and understory species and not to differences in projected spring temperature changes among

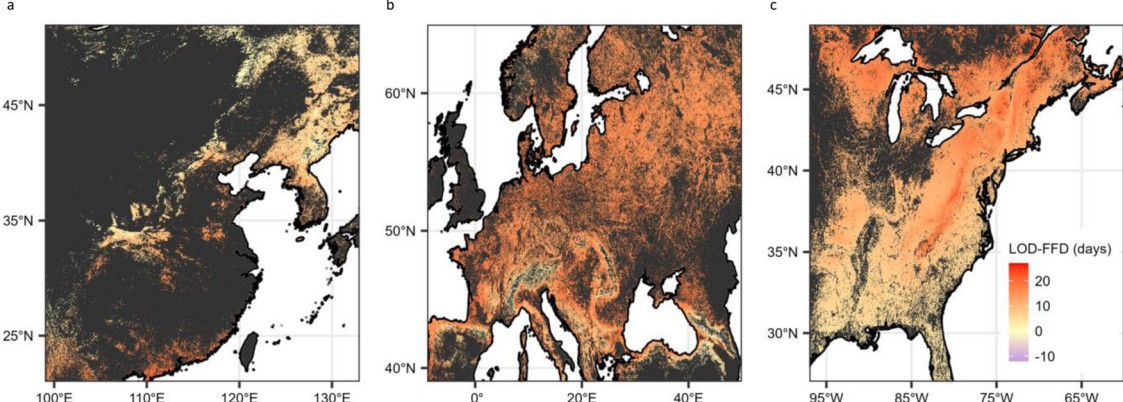

**Fig. 3 | Current estimated phenological escape duration in northern temperate deciduous forests.** Estimated mean difference between wildflower First Flower Date (FFD) and canopy tree Leaf Out Date (LOD) (in days) under current climate conditions (averaged from 2009–2018, see methods) in **a** Asia, **b** Europe, and **c** North America. Negative values indicate tree LOD is estimated to occur before wildflower FFD. Estimations were cropped by the estimated area of broadleaf and mixed-broadleaf forest (see methods). Dark gray regions indicate areas where the consensus land classification is <1% deciduous or mixed deciduous forest cover. An uncropped version of this figure is available in Fig. S11.

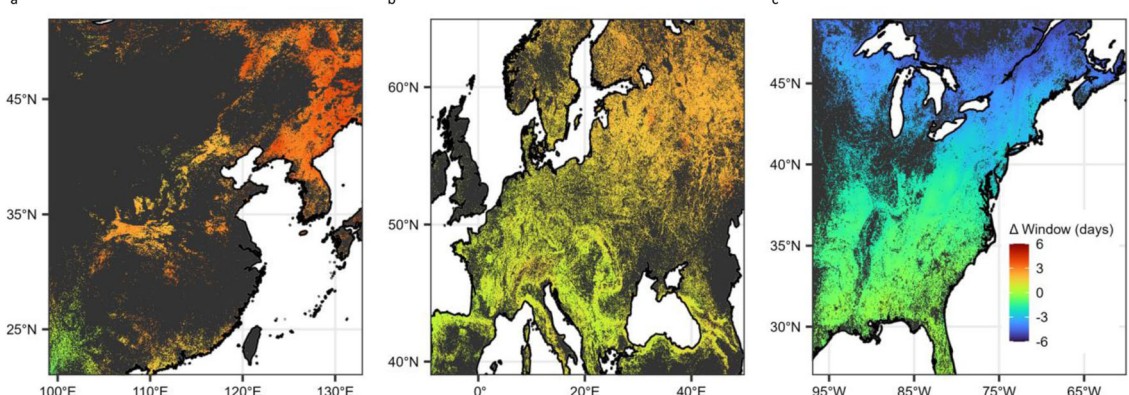

**Fig. 4 | Projected changes in phenological escape between current climate conditions and those projected for the end of this century.** Projected change in spring light window duration (ΔWindow, days) in **a** Asia, **b** Europe, and **c** North America. Positive values indicate regions where light window duration is expected to increase with spring warming (particularly in northern Asia) and negative values indicate where it is expected to decrease (particularly in northern North America). Dark gray regions indicate areas where the consensus land classification is <1% deciduous or mixed deciduous forest cover. An uncropped version of this figure is available in Fig. S12.

continents. Projected changes in March–April temperature broadly overlapped across continents (mean ± s.d. of projected change in spring temperature is 5.41 ± 1.87 °C, 6.36 ± 2.36 °C, and 5.04 ± 2.92 °C in Asia, Europe, and North America, respectively; Fig. S8). Intracontinental differences, however, such as the more extreme projected changes in phenological escape in the northern regions of each continent, are due to gradients of projected spring temperature change across latitudes, with higher latitude regions expected to experience greater spring warming relative to lower latitudes[34,35].

The North American results are consistent with findings from a previous study that similarly predicted reduction in access to spring light at a single site in the U.S. Northeast[7]. Although no previous studies have specifically addressed phenological escape in Asia, our results are consistent with previous research that found higher sensitivity to spring temperature for Asian herbaceous plants compared to Asian woody trees and shrubs[3]. Similarly, we are not aware of European studies that have directly compared phenological sensitivity between wildflowers and trees nor that have quantified phenological escape, but our estimates of sensitivity are consistent with previously reported values[8,9]. Together, this supports our conclusion that the duration of spring light windows will be differently affected by climate change on different continents.

Importantly, it is impossible to fully measure the ecological and demographic implications of these results using herbarium specimens. Linking changing access to light to changes in photosynthetic carbon assimilation, growth, and survival necessitates the use of experimental studies of living specimens such as those previously conducted in eastern North America[7,12,13,16]. Previous studies that have done so have linked access to spring light before canopy closure with the growth[10,12,17–19], survival[12,18–20], flowering[17,20], and reproductive output[10,17,21,22] of understory plant species. This large body of research suggests that reductions in access to spring light can have negative ramifications for the demographic success of understory species.

For example, projected reductions of several days in phenological escape may not intuitively seem like much, but the spring ephemeral wildflowers in this study have growing seasons that last only 3–5 weeks. A decrease in one day of phenological escape can therefore be as much as a 4% loss of total growing season length. In a study of phenological escape in sugar maple seedlings, Kwit et al.[14] found that increasing access to spring light by three days could increase photosynthetic carbon assimilation by more than 50%, an amount that other studies suggest would substantially increase annual growth and probability of survival[12,13]. For spring ephemeral herbaceous plants, Heberling et al.[16] showed that a similar three-day loss in access to

spring light resulted in up to a 30% loss in spring carbon assimilation. It is important to note, however, that the exact impacts of change in phenological escape on plant performance in Asia and Europe remains to be directly measured.

Thus, although our results suggest strong differences in change in performance when assuming the minimal differences in photosynthetic rates among species on different continents, it is possible that systematic intercontinental differences could lead to different outcomes. For example, if spring wildflowers were to be shown to have much lower spring photosynthetic rates, differences in projected access to light could have much less impact on overall carbon assimilation and demographic performance. Still, others have found relatively strong relationships between spring light availability and understory plant performance on other continents[21], suggesting that photosynthetic rates are similar enough to be comparable in this study. Therefore, we conclude that the relatively small changes in spring light availability projected here are likely to have substantial ecological consequences for the performance and demography of understory plant species.

### The future for spring ephemerals

The herbaceous layer accounts for more than 80% of plant species in temperate forests worldwide[36] and provides a critical role in the functioning of these ecosystems[37]. Understory wildflower species face many threats that include deer herbivory[38–40], habitat loss[41], pollinator declines[42] and mismatch[43,44], presence of nonnative, invasive plants[45,46], and nutrient pollution[47,48]. Furthermore, there is substantial evidence that many of these species are limited in their dispersal ability[41,49] and populations are likely unable to shift their distributions as rapidly as regional climates are predicted to warm. Our results show that mismatch between understory and overstory phenology, specifically in North America, is another concern for this already threatened group[36,50]. Reductions in spring light windows will likely lead to reduced carbon gains each year for spring forest wildflowers[7,16], which may ultimately lead to population decline as plant fitness declines. Therefore, we highlight the need for future conservation efforts to consider the impact of loss of spring light in restoration planning, with simultaneous studies of overstory and understory responses together[37]. Furthermore, this information may assist conservation practitioners in their decisions of where to focus management or conservation efforts based on projected changes to future forest ecosystems.

The patterns examined in this study point to distinctions in plant phenological responses among continents, but they cannot provide a mechanistic explanation for why these differences exist. Wildflowers and trees advance their phenology with warming spring temperatures, for example, but why are North American trees more responsive than co-occurring wildflower species to warming? Some speculate it is because perennial wildflowers overwinter underground whereas tree buds overwinter aboveground; trees may therefore be more affected by air temperature relative to wildflowers[10,11]. Although global soil and near-surface temperature data are recently becoming available[51], there are as yet no historical estimates of soil temperatures for use in herbarium studies. Thus, future empirical studies will be needed to directly test this theory. Similarly, what are the mechanisms that allow North American trees to be more sensitive to spring temperature than trees in Europe and Asia? Studies using dormant twigs, potted seedlings, and trees growing in botanical gardens are possible methods for investigating this. Common garden experiments with manipulated treatments may provide a more mechanistic understanding of our observations, including to rule out potentially confounding variables such as intercontinental differences in soil properties, precipitation, and photoperiod.

Other recent research highlights the growing need to assess the potential for nonlinear phenological responses to spring warming. As reviewed by Wolkovich et al.[52], future phenological responses to spring warming may be nonlinear owing to a number of reasons (including threshold responses to extreme environmental conditions and interactions among drivers besides spring forcing). Accounting for factors that might drive nonlinearities in plant phenological response to warming will necessitate controlled experiments[52,53], which was not possible with the observational herbarium dataset used in our study. We did not find evidence for nonlinear responses to spring temperature, but we acknowledge that experimental studies that can experimentally manipulate and test for the impacts of spring forcing, winter chilling, and photoperiodic effects on plant phenology could further illuminate how nonlinearity affects phenological escape. Also, nonlinear responses may become more evident in the future if extreme weather conditions occur more frequently.

Our study represents an early attempt to address phenological questions using herbarium specimens at the intercontinental level. Digitization of herbarium specimens and other historical data and citizen science programs like iNaturalist and iSpot are making data increasingly available and enabling new research[54]. Additional research that utilizes international herbaria and collaborations to synthesize these data at a global scale will aid in addressing pressing ecological and climate change questions.

In sum, our results indicate that spring ephemeral wildflowers are similarly sensitive to spring warming across temperate forests in Asia, Europe, and North America, but that deciduous canopy trees in North America are more sensitive to spring warming compared to trees on the other two continents. Furthermore, differences in tree leaf-out sensitivity mean that North American wildflowers are projected to experience decreasing access to spring light under climate change conditions, whereas those in Asia and Europe are unlikely to experience substantial gains or losses in spring light availability. These intercontinental differences suggest that climate change may adversely affect spring ephemerals in North America more than on the other two continents, and thus that North American wildflowers may be especially threatened by warming springs.

## Methods
### Phenological data
Herbarium specimens from North America (see Miller et al.[5]), East Asia, and Europe were evaluated for common species of deciduous overstory tree species and understory deciduous wildflowers, prioritizing congeneric species living on more than one continent when possible, and with genus and family in mind when not possible. Specimens were scored for either Leaf Out Date[25] (LOD, tree species) or First Flowering Date[26,31] (FFD, wildflower species), consistent with previous studies comparing phenology across forest strata[7,55]. This approach assumes that herbaceous species' flowering time is tightly correlated with the timing of leaf out for these species[7]. This approach is currently widely accepted to be the best we have to assess shifts in phenology using herbarium specimens[31], but see also Buonaiuto et al.[56]. In particular, we selected wildflower species that flowered and leafed out at approximately the same time in early spring.

Specifically, we searched digitized repositories of herbarium collections on each of the three target continents. Digitized Asian herbarium specimens were provided by the Chinese Virtual Herbarium (https://www.cvh.ac.cn/) and Chinese Field Herbarium (http://english.ib.cas.cn/). European specimens were located and collated using the Global Biodiversity Information Facility (gbif; https://www.gbif.org/). In North America, digitized specimens were collated from seven online repositories, further detailed in Miller et al.[5]. We conducted searches for each species independently, prioritizing spring ephemeral wildflowers and deciduous tree species with the highest number of available observations across most of the relevant

geographic range. We then filtered our initial searches by phenophase (scored by hand), making sure we only included observations that were flowering (for the wildflowers) or newly leafed out (trees). Next, we identified specimens that were missing information but for which digitized images of the specimen sheet were available. For these individuals, we entered missing phenological and georeferencing data when available (see below). Individuals with missing information and that either were not fully digitized or that did not include the necessary information on the digitized herbarium sheet were excluded from further study.

Specimens were georeferenced by researchers fluent in the languages in which the specimen data were collected in. Georeferencing was completed using the most precise geographic scale recorded at time of collection (i.e., prioritized by exact coordinates, town/locality, county). Specimens that did not identify location to at least county (in North America) or locality-level (in Asia and Europe) were excluded from this analysis, which is common practice in herbarium studies given the lack of precise geolocation data for some older specimens. Only specimens collected within each species' native range were used. Specimens collected prior to 1901 were excluded because climate estimates were not available prior to this year[57]. In total, after accounting for specimens that were excluded from our original search, we collected data for a total of 40 species (22 tree species and 18 herbaceous species; Table S1) consisting of 5,522 individual specimens. The complete dataset of herbarium specimens used in this analysis is freely available[58].

## Climate data

Climate data were extracted from the Climate Research Unit gridded Temperature Series (CRU TS) data set[57] v4.05 using the georeferenced location for each specimen. Spring temperature was calculated as the average of the March and April average monthly temperatures for the year and the location associated with each specimen. This metric is consistent with other studies which found this period to be important in cueing temperate plant phenology[5,7,13,55] and preliminary analysis indicated that this was the best spring temperature metric for modeling plant phenological responses in our dataset (see *Justification for March–April Temperature Window*). Daily weather records were not available for many years and many locations.

## Justification for March–April Temperature Window

Prior to conducting our analysis, we explored the correlation strengths between different spring temperature windows and observed phenology day of year (DOY). We made the decision to use a fixed average March–April temperature window for several reasons. First, we were limited in how we evaluated spring forcing effects by the type of data/climate estimates we were able to access. Due to the nature of the historical herbarium specimens we used in this study, we were limited to using historical average temperature estimates aggregated at the monthly level. We were, therefore, unable to use more nuanced modeling approaches such as moving windows or accumulation metrics like growing degree days (e.g., ref. 59).

Next, we evaluated correlation coefficients between different spring temperature windows (one-, two-, or three-month averages) from between February and May to determine which window led to the highest correlative strength with observed phenology. We determined that the March–April window had consistently high correlative strength for wildflower and tree phenology across all continents (Fig. S9), indicating that it was a robust predictor of spring phenology. This makes intuitive sense, as the vast majority (83%) of observations in our study occurred during or after DOY 96 (~March 15th) and nearly all observations (99%) occurring on or after DOY 60 (~March 1st; Fig. 1a).

Lastly, using a fixed March–April temperature window allows us to directly compare our results to other phenological escape studies using the same window[5,7,13].

## Models and analysis

We modeled the day of the observed phenological event (OPE) for individual $i$ of species $j$ using a normal likelihood distribution:

$$OPE_{i,j} \sim N(\mu_{i,j}, \sigma^2) \qquad (1)$$

The mean, μ, was modeled with an intercept term ($\beta 0$), slope terms representing phenological sensitivity to average spring temperature ($\beta 1$) and elevation ($\beta 2$), and species random effects ($\alpha_j$):

$$\mu_{i,j} = \beta 0 + \beta 1 \times SpringT_i + \beta 2 \times Elevation_i + \alpha_j \qquad (2)$$

We used slightly informative priors to estimate parameters: $\beta 0, \beta 1, \beta 2, \alpha_j \sim N(0, 1E\text{-}3); 1/\sigma^2 \sim Uniform(0,100)$. Other potential drivers of leaf-out and flowering phenology were explored in preliminary analysis (i.e., winter temperatures, annual precipitation, and spring precipitation), but these drivers did not generally improve model performance (Table S3) and were thus excluded from the model structure.

Models were run separately for each stratum (i.e., tree vs. herbaceous) x continent combination using the *R2jags* package[60] (v0.7-1) in R v4.1.0. Parameter values (means, variances, and covariances) were estimated from posterior distributions and are considered significantly different if the 95% Bayesian credible intervals (BCIs) of their posterior distributions do not overlap. Model code and data used to fit each model are publicly available[58].

## Description of spatial autocorrelation models

Spring leaf-out and flowering phenology have been extensively linked to many drivers, including spring and winter temperatures, interannual variability in temperature, photoperiod, the timing of snowmelt, precipitation, and elevation[2,5,7,9,12,61,62]. Importantly, many of these drivers covary along geographical gradients and particularly latitudinal gradients. Recent work published by Willems et al.[9] found that spatial autocorrelation among observations improved predictability of understory plant phenology in Europe, even when a host of other drivers were already accounted for. Their results thus suggest that there may be effects that are not commonly accounted for in phenology models associated with spatial proximity of the observations and that including spatial autocorrelation in such models could be important.

In this study, we first evaluated the fit of our phenological models using different combinations of potential drivers that included average spring temperature, average winter temperature, annual precipitation, spring precipitation, and elevation. We then assessed how incorporating spatial autocorrelation into the model structure affected model fit and posterior distributions of model parameters. To do so, we incorporated Integrated Nested Laplace Approximations (INLA) that include a spatial autocorrelation term into the model structure via the *inlabru* package[63] in R.

INLA modeling uses Stochastic Partial Differential Equations (SPDEs) as opposed to MCMC approximation methods[64] but has been demonstrated to maintain high fidelity to MCMC posterior estimates[65,66] and has the added advantage of allowing for the much faster estimation of spatial autocorrelative effects[64,66]. To account for the potential effects of including spatial autocorrelation on our results, we ran the original model structure using *inlabru* instead of *rjags* without and then with the spatial autocorrelation term added to it. Model fit and performance for *inlabru* models with and without spatial autocorrelation are located in Table S3. Comparison of parameter posterior estimates are located in Table S4. Matérn correlation functions were constructed in *inlabru* with a maximum edge length of 0.5 decimal degrees and a cutoff value of 0.25 decimal degrees. Resulting meshes can be found in Fig. S10.

A summary of results from this version of the analysis that includes spatial autocorrelation effects, as well as a detailed justification for having excluded them in the final analysis presented here, is included in the Supplementary Information.

## Climate Change Modeling

To forecast changes in the duration of spring light windows, we compared FFD and LOD from two different climate simulations. The first simulation represented current environmental conditions and was estimated by taking the average of spring (March–April) temperatures from a recent ten-year period (2009–2018). Climate and elevation data were downscaled from the CRU TS4.03[57] dataset using WorldClim 2.1[67] for bias correction at a resolution of 2.5 min. The average monthly temperature for each month and each year was calculated as the mean of monthly minimum and maximum temperatures.

The second simulation represents projected environmental conditions under climate change at the end of this century (2081–2100). Climate data used in this simulation were downscaled (using World-Clim 2.1[67]) from the Institut Pierre-Simon Laplace (IPSL) CM6A-LR climate model[34,68], which is part of the ongoing Climate Model Intercomparison Project[35] (CMIP6). We specifically used forecasts based on the Shared Socioeconomic Pathway (SSP) 585, which is analogous to the Representative Concentration Pathway (RCP) 8.5 from CMIP5 and earlier. This SSP is the most extreme, "business as usual", pathway used by the Intergovernmental Panel on Climate Change (IPCC). Therefore, projections made using this pathway represent the extreme threshold of climate change effects by the end of the century.

Model output is presented for regions where land-use is classified as > 1% temperate deciduous forest type according to a consensus of four global land-use models[1] (uncropped versions of figures can be found in the Supplementary Information as Figs. S11–12). Herbarium specimens were included in the above analyses regardless of the land-use classification of the respective georeferenced locations. We did not find consistent effects of land-use on phenological sensitivity, so we did not include this variable as a fixed effect in the final model. Still, we only present output for land areas currently covered by deciduous forests as those are so far the only systems where phenological escape has been shown to be important[5,7,10,12,13,16].

Approximations of FFD and LOD (for wildflowers and trees, respectively), were calculated using posterior mean estimates of the slopes, intercepts, and species-level random effects of our models (Tables S4-S7). Spring light window length was then calculated as the difference (in days) between FLD and LOD in each simulation X continent combination. Data processing and handling were completed primarily using the *ncdf4*[69], *stars*[70], and *sp*[71] packages in R v4.1.0.

**Inclusion and ethics statement.** The work presented here is the culmination of international collaboration and facilitation. Herbarium specimens were collated across three continents by researchers in those locations who spoke the language(s) in which information was recorded. All researchers who led these efforts are coauthors of this manuscript, and their employees and lab members who assisted in this effort are listed in the Acknowledgements section. Local and regional studies have been cited throughout this document, further acknowledging the contributions of global scholars to our work and to science as a whole.

## Reporting summary

Further information on research design is available in the Nature Portfolio Reporting Summary linked to this article.

## Data availability

The individual specimen phenology and environmental conditions data generated in this study have been deposited in the Zenodo database at https://doi.org/10.5281/zenodo.7080193. The processed spatial autocorrelation data are available in the same location. Past, current, and future climate estimates were acquired from the World-Clim 2.1 [https://www.worldclim.org/data/index.html] and CRU TS4.03 [https://crudata.uea.ac.uk/cru/data/hrg/] datasets.

## Code availability

Examples of code used in this analysis are freely available at https://doi.org/10.5281/zenodo.7080193.

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

## Acknowledgements

We thank A. Helmling, L. de Castro Machado de Souza, P.Y. Lai, C. Terry, and M. Rothendler (North American specimens); I. Vogler, M. Schramm, V. Bastian, K. Adler, and B. Müller (European specimens); and Z. Yang, C. Tan, L. Wang, and X. Huo (East Asian specimens) for their efforts in databasing and georeferencing the herbarium specimens analyzed for this project. We also thank C. Römermann, E. Welk, and B. Liu for their suggestions on the European and Asian species list. We are grateful to the Chinese Virtual Herbarium and Chinese Field Herbarium for providing the East Asian specimen data and digitized photos and to the *Metasequoia* funding of Nanjing Forestry University [YY], which helped fund the Asian data extraction. We thank F Lindgren and F Willems for providing guidance on INLA methodology for the models with spatial autocorrelation. BRL, TKM, JMH, SEK, and RBP were funded by National Science Foundation Grants Nos. DEB 1936971, 1936877, and 1936960.

We acknowledge the World Climate Research Program, which, through its Working Group on Coupled Modeling, coordinated and promoted CMIP6. We thank the climate modeling groups for producing and making available their model output, the Earth System Grid Federation (ESGF) for archiving the data and providing access, and the multiple funding agencies who support CMIP6 and ESGF. We thank Abe Miller-Rushing and Amanda S. Gallinat for providing valuable feedback on this manuscript.

## Author contributions

J.M.H., S.E.K., and R.B.P. initially conceived this project. T.K.M., C.R., and Y.Y. collected, scored, and georeferenced the herbarium collections used in this analysis. BRL conducted the statistical analysis and wrote the article. All authors contributed equally to reviewing and editing the manuscript.

## Competing interests

The authors declare no competing interests.
