## [Peer Review File · Nature Communications]

Reviewers' Comments:

Reviewer #1:

Remarks to the Author:

The authors use herbarium records as a source of data on wildflower flowering dates and tree leaf out dates, for Europe, Asia and North America. They then quantify the temperature sensitivity of these phenological events and examine how they vary across plant functional types (wildflowers vs trees) and continents, and what this means for climate change impacts on the herbaceous strategy of "phenological escape" and the duration of the "spring light window."

I think this is an important and valuable study, with clear ecological implications, but I have concerns about the way the statistics are presented and interpreted, and I am confused about what differences are statistically significant. See my comments below re: L107, 158, and 169.

And, while I agree with the statement on L87 that "Our results suggest that there are important differences in the sensitivity of tree species (but not wildflowers), with North American trees significantly more sensitive to spring temperature than trees on the other two continents," my concern about the temperature integration period (see comment re: L133) leads me to wonder if the reported differences in tree sensitivity to temperature across continents is an artifact of using the fixed March-April window. Indeed, it is worth noting that in the southern US, trees may leaf out in early March, and presumably the herbaceous layer is even earlier. So the fixed March-April window seems to me to be a poor choice.

Finally, the authors make many "recommendations" for future studies and advise how and what researchers "should" do. I am not convinced this guidance is useful or needed.

L107. I don't quite understand the "consistency" the authors point to, given that the CIs don't overlap for Asia. Similarly in L123, "generally consistent" seems to be an over-statement given that the previously reported values for Europe and North America are outside of the CIs reported in this study.

L129. I think claiming "predictive accuracy" of the models is misleading, because as far as I can tell this is based on a Bayesian model fit to the data (L458). There do not appear to be any independent data available for true validation. The authors should be more precise in their language, e.g. on L130 they should say " r^2 coefficients of fit vs observed phenology".

L133. I do not understand the assertion "difference in model performance could be due to the lower predictability of spring temperatures in North America." The explanation on the lines that follow does not make sense to me; it suggests to me that the temperature-driven model being used is wrong and that temperatures are being integrated over a longer period than the plants in Europe and Asia are sensitive to.

L158. The authors declare that estimates of temperature sensitivity are "considered significantly different from each other if confidence intervals do not overlap." But they claim "wildflowers [are] relatively more sensitive to spring temperature than trees in Asia and Europe but significantly less sensitive than trees in North America." Looking at Figure 1, I see that the CIs for wildflower FFD and tree LOD overlap in both Asia and Europe, so I am not sure I agree with the claim that wildflowers are "more sensitive" than trees to spring temperature. While the CIs for wildflower FFD and tree LOD are non-overlapping with each other for North America, they both appear to have some overlap with the European wildflower FFD, leading me to wonder about the claim that "relative difference between wildflower and tree sensitivity varied substantially among continents."

L169. Again, given the uncertainties reported here, it would appear that differences in the duration of the light window are also not different across continents?

Reviewer #2:

Remarks to the Author:

Review of "Wildflower phenological escape differs by continent and spring temperature"

Summary

The authors investigate "phenological escape" in North American, European, and Asian herbarium collections across the northern temperate deciduous forest biome. They find that phenological sensitivity to spring temperature is well conserved for understory wildflowers and that canopy trees in North America are significantly more sensitive to spring temperature than those found on the other continents.

General Comments

This is an interesting paper, which addresses an important topic in light of observed phenological shifts with climate change. The intercontinental comparison of phenological escape is interesting and valuable.

Major Concerns

However, I do have some major concerns with the manuscript as currently written:

- 1) For their forecasts of changes in phenological escape, the authors rely on simply linear relationships which are unlikely to remain consistent with warming. Phenological patterns are likely to follow nonlinear patterns in response to warming due to multiple cues shifting with climate change, threshold responses, and other factors (e.g., Wolkovich et al 2022). The authors need to discuss the implications of using only simple linear models focused on mean spring temperature (as they do), rather than process-based/physiological models (e.g. Chuine et al 2017) and/or models that incorporate other known drivers of spring phenology (e.g., chilling, photoperiod, Ettinger et al 2020). How would using some of these other common phenology models alter their results?
- 2) The authors need to give a bit more context for interpreting their findings, ideally in the main text. This includes providing the reader with an understanding of the mean spring temperature, and the mean and variation in day of year of phenological events (for LOD and FF), by continent and across species.
- 3) One of their main points, that there is intercontinental variation in phenological escape, seems to disappear when spatial autocorrelation is incorporated into their modelling. Could the authors discuss this more explicitly and clearly in the main manuscript?

Minor editorial suggestions:

Line 54: I suggest deleting "are" so that the phrase reads "trees because the former often overwinter underground..."

Line 72: The authors write: "It is similarly unknown what patterns for understory herbaceous species look like." I suggest being a bit more specific here, as it is unclear what "to patterns" the authors are referring (i.e., "it is similarly unknown what patterns of XXX for understory...")

Lines 124-125: I'm a bit baffled by the authors' comparisons of their findings for understory perennials with those from previous studies. They state that their estimates are "...generally consistent with those reported elsewhere in the literature (-2.2 days °C⁻¹ in North America and -3.6 days °C⁻¹ in Europe)." However their estimate is -3.14, [-3.28, -3.00] days °C⁻¹ in North America, about 30% stronger (nearly a full day greater shift per degree C) than those previous studies. Can the authors discuss why the estimates from their study are a greater magnitude for North America than previous studies? Perhaps it is a function of the focal taxa (especially since there is wide variation in temperature sensitivity across taxa, e.g. Vitasse et al 2009), ecosystems from which they were sampled, or other factors?

Line 185-187: The authors write "Dramatic differences in the projected change in spring light window duration emerged, ranging from increasing access to spring light for wildflowers in Asia, to minimal change in either direction in Europe, to decreasing light in North America..." Could the authors add a discussion of whether these differences are driven primarily by the differences in sensitivity across continents, the differences in forecasted shifts in temperature (and thus their forecasts of phenology), or both?

Line 469 and elsewhere: I believe the term "confidence intervals" is inappropriate for Bayesian analyses. Perhaps the authors mean "uncertainty intervals" or "credible intervals"?

Could the authors add plots and/or tables showing the temperatures, by continent? In interpreting their results, it would be helpful to know how the mean and variation in mean spring temperature differs across continents. This could affect their estimates of temperature sensitivity (e.g., Wolkovich et al 2021) and have other implications for interpreting their results.

References

Chaine I, Régnière J. Process-based models of phenology for plants and animals. *Annual Review of Ecology, Evolution, and Systematics*. 2017 Nov 2;48:159-82.

Ettinger AK, Chamberlain CJ, Morales-Castilla I, Buonaiuto DM, Flynn DF, Savas T, Samaha JA, Wolkovich EM. Winter temperatures predominate in spring phenological responses to warming. *Nature Climate Change*. 2020 Dec;10(12):1137-42.

Vitasse Y, Delzon S, Dufrêne E, Pontauiller JY, Louvet JM, Kremer A, Michalet R. Leaf phenology sensitivity to temperature in European trees: Do within-species populations exhibit similar responses?. *Agricultural and forest meteorology*. 2009 May 7;149(5):735-44.

Wolkovich EM, Auerbach J, Chamberlain CJ, Buonaiuto DM, Ettinger AK, Morales-Castilla I, Gelman A. A simple explanation for declining temperature sensitivity with warming. *Global Change Biology*. 2021 Oct;27(20):4947-9.

Wolkovich EM, Chamberlain CJ, Buonaiuto DM, Ettinger AK, Morales-Castilla I. Integrating experiments to predict interactive cue effects on spring phenology with warming. *New Phytologist*. 2022 May 22.

REVIEWER COMMENTS

NOTE: The line numbers we refer to in our responses refer to the line numbers when “All markup” is selected in the Track Changes menu in MS word.

Reviewer #1 (Remarks to the Author):

The authors use herbarium records as a source of data on wildflower flowering dates and tree leaf out dates, for Europe, Asia and North America. They then quantify the temperature sensitivity of these phenological events and examine how they vary across plant functional types (wildflowers vs trees) and continents, and what this means for climate change impacts on the herbaceous strategy of “phenological escape” and the duration of the “spring light window.”

I think this is an important and valuable study, with clear ecological implications, but I have concerns about the way the statistics are presented and interpreted, and I am confused about what differences are statistically significant. See my comments below re: L107, 158, and 169.

RESPONSE: With respect to our statistical approach and interpretation of statistical significance, we clarified our language and added more detail about our approach. We include details of our edits in our responses below.

And, while I agree with the statement on L87 that “Our results suggest that there are important differences in the sensitivity of tree species (but not wildflowers), with North American trees significantly more sensitive to spring temperature than trees on the other two continents,” my concern about the temperature integration period (see comment re: L133) leads me to wonder if the reported differences in tree sensitivity to temperature across continents is an artifact of using the fixed March-April window. Indeed, it is worth noting that in the southern US, trees may leaf out in early March, and presumably the herbaceous layer is even earlier. So the fixed March-April window seems to me to be a poor choice.

RESPONSE: The reviewer brings up an important point about the possibility that our use of average March-April spring temperatures may be inappropriate for some specimens collected at lower latitudes. Details and justification on why we chose this spring temperature window is now explicitly included in this revised text.

In our revised manuscript, we address the reviewer’s concern by adding a new figure to the main text (now Fig. 1) showing the distribution of observations in comparison to the spring light window we used for ease of visualization and details in how we selected our spring temperature variable in the Supplement (beginning on SL123). Importantly, nearly all observations (99%) occurred on or after DOY 60 (~March 1st), which demonstrates that our selected temperature window is appropriately timed for the vast majority of wildflower flowering and tree leaf out phenology dates. Furthermore, we found (see details below) that this temporal window was robust across our observations for both forest plant types and across continents.

We explored many potential temperature variables in preliminary data analysis, and we decided to use a fixed March-April window for several reasons:

- First, because of the broad spatial and temporal extent of our unique dataset, we were limited in the availability of climate data for our study. We were limited to using the

Climate Research Unit gridded Temperature Series (CRU-TS) dataset that provided us historical average monthly temperature estimates. We were thus unable to apply modeling approaches such as moving windows or accumulation metrics like growing degree days that require more detailed daily climate data.

- Second, we evaluated correlation coefficients between a suite of possible spring temperature windows to assess the temperature variable that was best-suited for our final model. We assessed one-, two-, and three-month average temperatures for all possible monthly estimates from February-May to determine which window led to the highest correlative strength with observed phenology. This analysis demonstrated that the March-April window had consistently high correlative strength for wildflower and tree phenology across all continents, indicating that it was a robust predictor of spring phenology for our final models (see new Fig. S11).
- Finally, to specifically address the reviewer’s concern of latitudinal bias in our spring temperature variable selection in North America, we conducted an additional analysis to test the predictive power of March-April average temperatures for observations binned by latitude (northern, central, and southern observations). We found that March-April temperature windows again provided either the best or among the best correlation strength between spring temperature and observed phenology across northern, central, and southern regions.

Therefore, our preliminary analysis of the best temperature variable for our models, and the fact that the majority of our phenological observations occur during or immediately following the March-April window, suggest that March-April temperatures are the most appropriate and a

robust predictor of spring phenology across continents, plant forms, and latitudes. We have summarized this justification in a new supplemental section on SL123.

Finally, the authors make many “recommendations” for future studies and advise how and what researchers “should” do. I am not convinced this guidance is useful or needed.

RESPONSE: In our revised manuscript, we removed the word “strongly” and added “consider” on L262 to mitigate the tone of telling other researchers what they should do. We also removed the paragraph beginning L292, which could be argued as being too speculative as it refers more to ecophysiological research and is less derived from our results.

L107. I don’t quite understand the “consistency” the authors point to, given that the CIs don’t overlap for Asia.

RESPONSE: We changed our framing of the comparison of our results to previous research in this paragraph, clarified that by “consistency” we are referring to overlap in 95% Bayesian credible intervals when applicable, and have given additional context for comparisons between herbarium-derived sensitivity and that derived from other methods. The section with our edits now begins on L122.

Similarly in L123, “generally consistent” seems to be an over-statement given that the previously reported values for Europe and North America are outside of the CIs reported in this study.

RESPONSE: We previously omitted the CIs of the parameter estimates that we cite from past literature, which we now include beginning on L143. There is broad overlap in the estimated Bayesian credible intervals of the phenological sensitivity in our study and in previous studies.

L129. I think claiming “predictive accuracy” of the models is misleading, because as far as I can tell this is based on a Bayesian model fit to the data (L458). There do not appear to be any independent data available for true validation. The authors should be more precise in their language, e.g. on L130 they should say “ r^2 coefficients of fit vs observed phenology”.

RESPONSE: We clarified our language in this sentence starting on L152, which now reads:

“Particularly noticeable in our results was that r^2 coefficients of predicted versus observed phenology were much higher in North America (0.70 and 0.76 for wildflower and tree models, respectively) compared to Asian (0.40 and 0.44, respectively) and European models (0.41 and 0.25, respectively).”

L133. I do not understand the assertion “difference in model performance could be due to the lower predictability of spring temperatures in North America.” The explanation on the lines that follow does not make sense to me; it suggests to me that the temperature-driven model being used is wrong and that temperatures are being integrated over a longer period than the plants in Europe and Asia are sensitive to.

RESPONSE: We edited this paragraph to clarify our assertion. Prior research on North American phenology has demonstrated that North American springs have higher interannual variability in spring temperatures relative to Europe or Asia (Zohner and Renner 2017). In other words, North American springs are marked by wide temperature fluxes of periods of high spring temperatures followed by abrupt cold temperatures. This high intra-annual variability in spring temperature could explain the differences in tree LOD phenological sensitivity we found between trees from the three continents. To clarify our text here, we added a sentence at the end of this paragraph clarifying that different sensitive periods are possible, but that we do not find this to be likely in this case because of our justification for use of the March-April window (see response to Rev 1 L87 criticism above). This section now starts on L156.

L158. The authors declare that estimates of temperature sensitivity are “considered significantly different from each other if confidence intervals do not overlap.” But they claim “wildflowers [are] relatively more sensitive to spring temperature than trees in Asia and Europe but significantly less sensitive than trees in North America.” Looking at Figure 1, I see that the CIs for wildflower FFD and tree LOD overlap in both Asia and Europe, so I am not sure I agree with the claim that wildflowers are “more sensitive” than trees to spring temperature. While the CIs for wildflower FFD and tree LOD are non-overlapping with each other for North America, they both appear to have some overlap with the European wildflower FFD, leading me to wonder about the claim that “relative difference between wildflower and tree sensitivity varied substantially among continents.”

REPOSENSE: We changed our wording on L186 to reflect that there is no statistical difference in sensitivity between wildflower and tree sensitivity in Asia or Europe.

L169. Again, given the uncertainties reported here, it would appear that differences in the duration of the light window are also not different across continents?

REPOSENSE: We changed our phrasing to reflect that there is broad overlap in the range of observed light window duration among the three continents studies, specifically referencing overlap in 95% credible intervals. This section now starts on L196.

Reviewer #2 (Remarks to the Author):

Review of “Wildflower phenological escape differs by continent and spring temperature”

Summary

The authors investigate “phenological escape” in North American, European, and Asian herbarium collections across the northern temperate deciduous forest biome. They find that phenological sensitivity to spring temperature is well conserved for understory wildflowers and that canopy trees in North America are significantly more sensitive to spring temperature than those found on the other continents.

General Comments

This is an interesting paper, which addresses an important topic in light of observed phenological shifts with climate change. The intercontinental comparison of phenological escape is interesting and valuable.

RESPONSE: Thank you.

Major Concerns

However, I do have some major concerns with the manuscript as currently written:

- 1) For their forecasts of changes in phenological escape, the authors rely on simply linear relationships which are unlikely to remain consistent with warming. Phenological patterns are likely to follow nonlinear patterns in response to warming due to multiple cues shifting with climate change, threshold responses, and other factors (e.g., Wolkovich et al 2022). The authors need to discuss the implications of using only simple linear models focused on mean spring temperature (as they do), rather than process-based/physiological models (e.g. Chuine et al 2017) and/or models that incorporate other known drivers of spring phenology (e.g., chilling, photoperiod, Ettinger et al 2020). How would using some of these other common phenology models alter their results?

RESPONSE: We agree that there is a robust body of evidence suggesting that nonlinearities between plant phenology and warming can arise under certain conditions, as outlined in the Wolkovich et al. 2022 review. However, the trends that we found were strongly linear (see figure below) and did not suggest that there were nonlinear trends or threshold responses we were missing.

Furthermore, we evaluated the possible contribution of winter chilling effects to phenology (see L374 in methods and table S3), but they did not improve model performance and we therefore excluded winter chilling variables from the final analysis. Thus, in our study, we did not find evidence of interacting temperature drivers or nonlinear responses to phenology.

Still, we added a new paragraph to our discussion section (beginning L281) that describes the possibility of nonlinearities and the importance of future studies to empirically test how multiple phenological cues interact to predict plant phenological response to changing spring temperatures.

This figure shows the relationships between spring (March-April) temperature and observed phenology for wildflowers (red, top) and trees (blue, bottom), in Asia (left), Europe (middle), and North America (right). Dashed lines indicate relationships fit to a linear relationship whereas solid lines indicate relationships fit to a loess curve (i.e., allowing for nonlinearity in trends). Linear and nonlinear models of Asian and European data had complete overlap of standard error bands. North American linear and nonlinear models only diverged near the tails of distributions where there was less data available.

2) The authors need to give a bit more context for interpreting their findings, ideally in the main text. This includes providing the reader with an understanding of the mean spring temperature, and the mean and variation in day of year of phenological events (for LOD and FF), by continent and across species.

RESPONSE: As suggested by reviewers 1 and 2, we added more context for justifying and explaining our use of mean spring temperature. We clarified that we used average March-April temperature on L89 of this manuscript and a more detailed justification of our selection procedure for this variable on SL123 of the supplement. Furthermore, we added a new graph (now Fig. 1, see response to Reviewer 1 above) showing the distribution of observed phenology used in this study, and a new supplemental table (Table S2) listing the means and standard deviations of observed DOY and estimated spring temperatures. We added a sentence beginning on L90 with a brief justification of why we chose this spring temperature window.

3) One of their main points, that there is intercontinental variation in phenological escape, seems to disappear when spatial autocorrelation is incorporated into their modelling. Could the authors discuss this more explicitly and clearly in the main manuscript?

RESPONSE: We include a brief discussion of the impacts of including a spatial autocorrelation parameter in the paragraph beginning on L171. We also discuss in more detail why we choose to leave the spatial autocorrelation parameter out of the models. Briefly, it is because of large inconsistencies in spatial autocorrelative effects, particularly in Asia (effects of autocorrelation adjusted phenology estimates +/- 50 days). We found evidence that spatial autocorrelation estimates were partially accounting for the impact of elevation on phenology, and by including elevation in our main models, we are accounting for a major source of spatial autocorrelation in the dataset. We also include a more detailed assessment of the impacts of spatial autocorrelation term in the supplement materials (SL7).

Minor editorial suggestions:

Line 54: I suggest deleting “are” so that the phrase reads “trees because the former often overwinter underground...”

RESPONSE: Changed wording

Line 72: The authors write: “It is similarly unknown what patterns for understory herbaceous species look like.” I suggest being a bit more specific here, as it is unclear what “to patterns” the authors are referring (i.e., “it is similarly unknown what patterns of XXX for understory...”

RESPONSE: We reworded this sentence to now read: “It is similarly unknown how understory plant phenological sensitivity varies across large spatial and temporal scales.”

Lines 124-125: I’m a bit baffled by the authors’ comparisons of their findings for understory perennials with those from previous studies. They state that their estimates are “...generally consistent with those reported elsewhere in the literature (-2.2 days °C-1 in North America and -3.6 days °C-1 in Europe).” However their estimate is -3.14, [-3.28, -3.00] days °C-1 in North America, about 30% stronger (nearly a full day greater shift per degree C) than those previous studies. Can the authors discuss why the estimates from their study are a greater magnitude for North America than previous studies? Perhaps it is a function of the focal taxa (especially since there is wide variation in temperature sensitivity across taxa, e.g. Vitasse et al 2009), ecosystems from which they were sampled, or other factors?

RESPONSE: As noted in our response to Reviewer 1’s comment on L123 above, we added the reported CIs that we compare our results to and there is broad overlap in CIs with our results, suggesting general consistency. In response to Reviewer 2, we have also added a sentence on L149 reading:

“Discrepancies in mean responses between this study and others may be due in part to different types of data (herbarium specimens versus field observations) and to choice in focal taxa, as temperature sensitivity has been shown to vary widely across taxa (Vitasse et al 2009).”

Line 185-187: The authors write “Dramatic differences in the projected change in spring light window duration emerged, ranging from increasing access to spring light for wildflowers in

Asia, to minimal change in either direction in Europe, to decreasing light in North America...”
Could the authors add a discussion of whether these differences are driven primarily by the differences in sensitivity across continents, the differences in forecasted shifts in temperature (and thus their forecasts of phenology), or both?

RESPONSE: We added a few sentences discussing this topic beginning on L222.

Line 469 and elsewhere: I believe the term “confidence intervals” is inappropriate for Bayesian analyses. Perhaps the authors mean “uncertainty intervals” or “credible intervals”?

RESPONSE: We replaced the term “confidence intervals” with “credible intervals” throughout the manuscript and in the supplementary material

Could the authors add plots and/or tables showing the temperatures, by continent? In interpreting their results, it would be helpful to know how the mean and variation in mean spring temperature differs across continents. This could affect their estimates of temperature sensitivity (e.g., Wolkovich et al 2021) and have other implications for interpreting their results.

RESPONSE: Fig. S1 includes a map showing the average current spring temperature across all three continents, but we had not added a representation of the variation in March-April temperature associated with the observed phenology. We now include a panel in our new Fig. 1 (panel b, see response to Reviewer 1 above), illustrating the distributions of spring temperature across the different continents and separated by plant group. We also added reference to Fig 1b on L108, which now reads:

“Following model selection, our final model structure included fixed effects of average spring (March-April) temperature, and elevation as well as species-level random effects. We show continental distributions of spring temperature values in Fig. 1b (means and standard deviations are listed in Table S2).”

Reviewers' Comments:

Reviewer #1:

Remarks to the Author:

Review of NCOMMS-22-28830

The clarity of the paper has been greatly improved in response to the first round reviews. But, as I read the revised paper I have new concerns about the results and the interpretation, which I list below.

L35. It seems to me that "advancing tree phenology will lead to decreasing spring light windows" is really a prediction that emerges from the results; this can be tested in the future. But it is far from guaranteed that this is what will actually happen, so the way it is worded here might be less definitive or assertive.

L49. I agree with the idea that herbaceous plants wintering underground might not be sensitive to air temperature to the same degree that trees are. But this would seem to me to then lead to a very conservative phenological strategy, which is less opportunistic than the underlying idea of "phenological escape." Any thoughts? And if this were the case, then I still don't understand why herbaceous plant flowering dates would be more sensitive to air temperature than tree leaf-out dates in Asia and Europe (Figure 2). This just seems counter-intuitive. It also raises questions about why the analysis is conducted using air temperature rather than soil temperature, at least for the herbaceous species.

L81. I understand the arguments the authors present for using mean March-April temperatures in their analysis, but this ultimately means that the effects of warming on phenology of all species are going to be driven (linearly) by how temperatures in those particular months might change, and I'm still not sure I find that credible. The idea that "all species respond the same, but with different sensitivities" isn't all that convincing. It doesn't give anyone much of a competitive edge, does it? Maybe I am misunderstanding.

L86. The argument about daily records not being available is also not all that convincing; the authors could do the analysis using daily records where those are available, and report whether (or where) the results are consistent with the monthly analysis. Since many plant processes are nonlinear with temperature, I suspect that the inference from monthly means would be different from using daily max/min data. I'm not convinced that just because a hierarchical Bayesian model (which a lot of readers won't understand) is used means that we can ignore the implicit weakness of using monthly data. And then there is the point I made above about air vs. soil temperature.

L120+. I find the claims here of "firsts" to be wearying. Is this really necessary?

L129+. With the uncertainties now clarified, are the differences between herbaceous species and trees really as important as the authors claim? For North America, it looks to be about half a day per degree, looking at the mean sensitivities. This makes me think that a deeper analysis, looking at the full life cycle, carbon balance, and reproductive output of the understory plants would really be needed to conclude that these differences in temperature sensitivity are ecologically meaningful (in other words, how close to the break-even point are understory plants under the climate regime they currently experience? If there is a decent safety margin, the narrowing of the spring light window may not be ecologically meaningful—the authors have not made the case that this is true). I'm just a bit skeptical given that the results for Europe and Asia are so different. Maybe the safety margin is already different in Europe and Asia vs. North America?

L256. The authors rejected previous criticism about the potential for nonlinear responses to warming, but then point to this possibility. I find this paragraph quite unconvincing.

L270. Rather than finishing by encouraging other researchers to follow along and replicate this analysis more broadly, I think a more compelling conclusion would highlight the main take-home messages and place them in a broader context. I really don't like it when I am told by a paper what I should do next.

Reviewer #2:

Remarks to the Author:

The authors have mostly addressed my previous concerns. I do have a few additional comments and questions:

Figure 2: Given the Bayesian approach, I suggest avoiding terms such as “statistically significant” which is grounded in frequentist approaches. I suggest rewording this phrase to say “Estimates are considered different from 0 if credible intervals do not overlap the dashed 0 line and are considered different from each other if credible intervals do not overlap.”

Line 130 Similarly, I suggest replacin “no statistically significant differences” with “no strong differences”

Line 150-154: Isn't an additional possibility that a different window of spring temperatures might result in higher correlation for Europe and/or Asia?

REVIEWER COMMENTS

Reviewer #1 (Remarks to the Author):

Review of NCOMMS-22-28830

The clarity of the paper has been greatly improved in response to the first round reviews. But, as I read the revised paper I have new concerns about the results and the interpretation, which I list below.

L35. It seems to me that “advancing tree phenology will lead to decreasing spring light windows” is really a prediction that emerges from the results; this can be tested in the future. But it is far from guaranteed that this is what will actually happen, so the way it is worded here might be less definitive or assertive.

RESPONSE: Thank you for the suggestion. We have changed the language in this sentence (on L34) from “Our findings reveal” to “We predict” to emphasize that these are predictions based on model results and not predetermination of future events.

L49. I agree with the idea that herbaceous plants wintering underground might not be sensitive to air temperature to the same degree that trees are. But this would seem to me to then lead to a very conservative phenological strategy, which is less opportunistic than the underlying idea of “phenological escape.” Any thoughts?

RESPONSE: The reviewer makes a good point here, and one that is an important implication for our work. Many wildflowers follow a phenological escape strategy, but in some instances they may be caught in an ecological trap that prevents them from responding as rapidly as trees to a changing climate. The strategy must balance advantages of early light access with frost risk potential. We make this point now more clearly in the discussion (L287).

L49 (continued): And if this were the case, then I still don’t understand why herbaceous plant flowering dates would be more sensitive to air temperature than tree leaf-out dates in Asia and Europe (Figure 2). This just seems counter-intuitive. It also raises questions about why the analysis is conducted using air temperature rather than soil temperature, at least for the herbaceous species.

RESPONSE: While posterior means may be in this direction, there is no statistically significant difference between wildflower and tree sensitivity in either Asia or Europe (i.e., the credible intervals overlap). To prevent this confusion in interpretation, we have adjusted the wording in the Results (L133, and throughout the paper), also consistent with suggestions from Reviewer 2.

Thank you for the suggestion of using soil temperature instead of air temperature. The reason that we use air temperature is that it is widely available across the three continents, whereas soil temperatures are not available across the broad spatial and temporal scales that our herbarium collections span. However, empirical studies demonstrate a strong positive correlation between air and soil temperatures generally (Zhan et al. 2019, *Advances in Meteorology*), and importantly

in temperate forest biomes during the spring (March through May; Qian et al. 2011, *Journal of Geophysical Research*). This builds confidence that our use of readily available air temperature data for predicting the phenology of trees and herbs, which may indeed be responding to air and soil temperature cues, respectively. It is an interesting and credible hypothesis that wildflower phenology should be more responsive to soil temperatures rather than air temperatures, and we include this hypothesis in our discussion. For now, there is no feasible way to include this analysis in our project. We call for future studies on more recent datasets to disentangle the relative importance of soil vs. air temperatures on herbaceous plant phenology in temperate ecosystems.

L81. I understand the arguments the authors present for using mean March-April temperatures in their analysis, but this ultimately means that the effects of warming on phenology of all species are going to be driven (linearly) by how temperatures in those particular months might change, and I'm still not sure I find that credible. The idea that "all species respond the same, but with different sensitivities" isn't all that convincing. It doesn't give anyone much of a competitive edge, does it? Maybe I am misunderstanding.

RESPONSE: When we say that all species are responding the same, we are indicating that all species become active earlier with warmer temperatures. However, different species have different sensitivities to temperature, and it is these differences that result in different advantages or competitive edges (Lee and Ibanez 2021 *Glob Change Biol*, Lee and Ibanez 2021 *Functional Ecol*, Heberling et al. 2019 *Ecol Letters*, Heberling et al. 2019 *New Phytologist*, Ge et al. 2015 *Global Change Biol*, etc.). Our hierarchical modeling structure allows for species-level differences in responses. Regarding your comment about linearity of responses, we carried out a careful and thorough analysis of the data, detailed in depth in the previous round of edits, and found that, in all cases, linear models were as good or better than curvilinear models in explaining plant phenology in relation to temperature (as detailed in our response to first-round reviews).

L86. The argument about daily records not being available is also not all that convincing; the authors could do the analysis using daily records where those are available, and report whether (or where) the results are consistent with the monthly analysis. Since many plant processes are nonlinear with temperature, I suspect that the inference from monthly means would be different from using daily max/min data. I'm not convinced that just because a hierarchical Bayesian model (which a lot of readers won't understand) is used means that we can ignore the implicit weakness of using monthly data. And then there is the point I made above about air vs. soil temperature.

RESPONSE: We agree with the Reviewer that it would be a useful exercise to compare the use of monthly average temperatures – which is currently the common method in these historical herbarium studies – to daily temperature min/max values to understand how aggregation of monthly temperature data impacts results and understanding. However, daily weather records are not available across the spatial or temporal resolutions of observations in our dataset to make use of any meaningful subset of the data to test these ideas. For example, even in Boston, Massachusetts, where we have a high density of herbarium records, daily weather station data is sparse and inconsistent across weather monitoring stations through time (see NOAA Climate

Data Online, www.ncei.noaa.gov/). Stitching together daily temperature records from weather station daily data across the greater Boston region would result in a different form of aggregated temperatures that would introduce different forms of variability and error associated with this form of data aggregation. Furthermore, weather station data are not publicly nor academically available in China, where more than 95% of our Asian herbarium specimens were collected, preventing us from being able to make the important intercontinental comparisons that are central to our thesis.

Using monthly temperature means, we created models with high R² values suggesting that our use of monthly mean data was capturing a large amount of variation in the phenological responses of the species we studied. This point is further supported by recent evidence showing that using this approach to model phenology from herbarium records is consistent with modern citizen-science-based estimates of phenology (Ramirez-Parada et al 2022, *Ecography*). The large amounts of variability and error associated with daily temperature records will likely not significantly improve model fit. Further, the use of monthly means is widely accepted in similar published herbarium-based studies (Ramirez-Parada et al. 2022 *Ecography*, Park et al. 2019 *Phil Trans B*, Davis et al. 2015 *Am J Botany*, Love and Mazer 2021 *Am J Botany*, Everill et al. 2014 *Am J Botany*, Primack et al. 2004 *Am J Botany*, Miller et al. 2021 *Annals of Botany*, Willis et al 2017 *Trends in Ecology and Evolution*, etc.).

Please see our response, above, justifying our use of linear models and air temperature data.

L120+. I find the claims here of “firsts” to be wearying. Is this really necessary?

RESPONSE: Thank you for pointing this out. We have changed the wording (L125).

L129+. With the uncertainties now clarified, are the differences between herbaceous species and trees really as important as the authors claim? For North America, it looks to be about half a day per degree, looking at the mean sensitivities. This makes me think that a deeper analysis, looking at the full life cycle, carbon balance, and reproductive output of the understory plants would really be needed to conclude that these differences in temperature sensitivity are ecologically meaningful (in other words, how close to the break-even point are understory plants under the climate regime the currently experience? If there is a decent safety margin, the narrowing of the spring light window may not be ecologically meaningful—the authors have not made the case that this is true). I’m just a bit skeptical given that the results for Europe and Asia are so different. Maybe the safety margin is already different in Europe and Asia vs. North America?

RESPONSE: Thank you for these suggestions. Some of the authors of this study have used this type of ‘life cycle’ or ‘carbon budget’ analysis to demonstrate how access to spring light is directly related to understory plant photosynthetic rates, carbon assimilation, and demographic performance (Lee and Ibanez 2021a and 2021b; Heberling et al. 2019a and 2019b). These carbon budgets estimate that wildflowers have lost ~1-3% of their mean annual carbon budget per day of light loss owing to shortening of spring light windows (Heberling et al. 2019a,b). While a few days may seem inconsequential for some ecological processes, if you consider that, for the species we chose, their growing seasons are only 3-5 weeks long, and one day can be as much as 4% of their total growing season length. Combined with our previous empirical results, our

current results suggest that these differences have important ecological implications. Given our previous work in this area, we added to the discussion (L226 and L241) these estimates of how seemingly small changes in access to light can lead to physiological implications for understory plant carbon budgets in both understory tree seedlings and herbaceous wildflowers.

We agree with the reviewer that an empirical analysis looking at the projected impact on reduced spring light, carbon budgets, and total fitness through time would be necessary to demonstrate that these phenological mismatches will lead to long-term changes in population size and fitness. While that analysis is outside the scope of our study, as it would require tracking living trees and wildflowers, not preserved herbarium specimens, we certainly hope that our findings will stimulate more research and publications in this area.

L256. The authors rejected previous criticism about the potential for nonlinear responses to warming, but then point to this possibility. I find this paragraph quite unconvincing.

RESPONSE: Based on comments in the previous round of reviews, we considered the possibility of nonlinear responses to temperature. We did exactly as the reviewer requested, and we found that linear models performed as well as or better than non-linear models. We found no evidence that species were reaching an asymptote or changing their sensitivity at extreme temperatures. In other words, we appreciated the suggestion from the last review and did not reject the criticism; we just did not find any evidence of non-linearity in the data. In response to the comments of the reviewer from the previous round of reviews, we did add in a paragraph considering how nonlinear responses might affect phenology in the future (L296).

L270. Rather than finishing by encouraging other researchers to follow along and replicate this analysis more broadly, I think a more compelling conclusion would highlight the main take-home messages and place them in a broader context. I really don't like it when I am told by a paper what I should do next.

RESPONSE: Thanks for these suggestions. Our goal in the conclusion was to suggest important next steps in this line of research. We have retained most of these points and, at the suggestion of reviewer 1, at the end of the conclusion, we add a new paragraph to place the take-home messages in a broader context (L314).

Reviewer #2 (Remarks to the Author):

The authors have mostly addressed my previous concerns. I do have a few additional comments and questions:

Figure 2: Given the Bayesian approach, I suggest avoiding terms such as “statistically significant” which is grounded in frequentist approaches. I suggest rewording this phrase to say “Estimates are considered different from 0 if credible intervals do not overlap the dashed 0 line and are considered different from each other if credible intervals do not overlap.”

RESPONSE: Thank you for this suggestion. We edited the caption to Fig. 2 (L639) to the suggested language.

Line 130 Similarly, I suggest replacing “no statistically significant differences” with “no strong differences”

RESPONSE: We made this change on L133

Line 150-154: Isn't an additional possibility that a different window of spring temperatures might result in higher correlation for Europe and/or Asia?

RESPONSE: We edited that sentence (L155) to clarify that alternative spring forcing windows are an important possibility in this regard.